# The importance of Atmospheric Correction for Airborne Hyperspectral Remote Sensing of Shallow Waters. Application to Depth Estimation.

Elena Castillo-López[1], Jose Antonio Dominguez[2], Raúl Pereda[1], Julio Manuel de Luis[1], Ruben Pérez[1], Felipe Piña[1].

[1] Department of Geographic Engineering and Techniques of Graphical Expression, University of Cantabria, 39005, Spain. Phone: +34 660738989. Fax number: +34 942201703; castille@unican.es.
[2] Department of Mathematical Physics and Fluids, Science Faculty, National Distance Education University (UNED).

*Correspondence to*: Elena Castillo-López (castille@unican.es)

**Abstract.** Accurate determination of water depth is indispensable in multiple aspects of civil engineering (dock construction, dikes, submarines outfalls, trench control, etc.). According to the final objective, different accuracies will be required. Accuracy in bathymetric information is highly dependent on the atmospheric correction made to the imagery. The reduction of effects such as glint and cross track illumination in homogeneous shallow-water areas ocimproves the results of the depth estimations. The aim of this work is to assess the best atmospheric correction method for the estimation of depth in shallow waters, considering that reflectance values cannot be greater than 1.5% because otherwise the background would not be seen. This paper addresses the use of hyperspectral imagery to quantitative bathymetric mapping, and explores one of the most common problems when attempting to extract depth information in conditions of variable water types and bottom reflectances. The current work assesses the accuracy of some classical bathymetric algorithms (Polcyn-Lyzenga, Philpot, Benny-Dawson, Hamilton, Principal Component Analysis) when four different atmospheric correction methods are applied and water depth is derived. None atmospheric correction is valid for all type of coastal waters but in heterogeneous shallow water, the model of atmospheric correction 6S offers good results.

## 1 Introduction

Coastal development activities alter coastal catchments and directly affect littoral environments. Management of these ecosystems requires improved monitoring systems to track changes in water quality and quantity through time, but such records are better contextualized by using synoptic data (Mertes et al., 2004), as this type of impacts are now commonly observed at region rather than local scale. Monitoring marine systems has always been difficult and expensive: many of these impacts have simply gone unrecorded.

Traditional in situ survey methods, such as bathymetries made with Global Positioning System (GPS) in Real Time Kinematic (RTK) mode and echo-sounder data (Pereda et al., 2016), reach higher accuracies and provide excellent data nowadays, but

they require major logistical commitments and often lack spatial-temporal resolution to resolve the aimed processes. While they perhaps provide lower accuracy, remote sensing techniques offer the potential for cost-efficient, long-term data collection with high resolution in time and space.

In this sense, the study of the water optics has historically been considered as the best alternative for depth estimation, and the behavior of light through the water column has been explained by water-leaving reflectance $R(0_+)$.

The relation between water-leaving reflectance just above the surface $R(0_+)$ and water-leaving reflectance just below the surface, $R(0_-)$, is 0.544 (Austin, 1974; Kirk, 1994; Kutser, 2004).

Water-leaving reflectance just below the surface, $R(\lambda, 0_-)$, is composed by two parts: the water volume reflectance and the bottom reflectance light (Philpot, 1987):

$$R_u(\lambda, 0_-) = R_w(\lambda, 0_-) + R_b(\lambda, 0_-)$$

Where $R_u(\lambda, 0_-)$ is the total spectral water-leaving reflectance seen, $R_w(\lambda, 0_-)$ is the total spectral water-leaving reflectance due to water, and $R_b(\lambda, 0_-)$ is the total spectral water-leaving reflectance influence of the bottom.

The bottom influence was determined by using a simple model:

$$R_b(\lambda, 0_-) = [R_b(\lambda, 0_-) - R_w(\lambda, 0_-)]e^{[(K_d \pm K_u)Z]}$$

Being $K_u$ the upwelling diffuse attenuation coefficient, $K_d$ the downwelling diffuse attenuation coefficient and Z the depth.

Considering the same downwelling and upwelling irradiance, $K_u = \pm K_d$ (Martitorena et al., 1994), (Dierssen et al., 2003), the following expression can be applied to study the bottom reflectance:

$$R(\lambda, 0_-) = R_w(\lambda, 0_-) + \left[\left(R_b(\lambda, 0_-) - R_w(\lambda, 0_-)\right)e^{[(2K_d)Z]}\right]$$

The water-leaving reflectance above water, $R(\lambda, 0_+)$, is related to the water column, and it is influenced by the bottom shape

$$R(\lambda, 0_+) = 0.544\left\{R_w(\lambda, 0_-) + [R_b(\lambda, 0_-) - R_w(\lambda, 0_-)]e^{[(-2K_d)Z]}\right\}$$

Nevertheless, the monitoring of coastal waters represents a challenging task due to their complexity (Morel and Prieur, 1977), (Gordon and Morel, 1983).

Since the 1980s, the development of remote sensing has permitted monitoring coastal waters. However, the use of this technology has added a new problem to solve: the influence of the atmosphere between the water and the sensor that is installed

in its platform. The correction of the atmospheric problem must analyze various contributions to the top-of-atmosphere ($TOA$) radiance measured by a satellite-borne sensor ($L_t$), such as the solar scattering by the atmosphere molecules and aerosols ($L_{atm}$), Sun and sky radiance reflected by the sea surface (either by the water surface itself or by foam from whitecaps) ($L_{TOA\_surf}$) and, finally, the water-leaving radiance ($L_{TOA\_w}$) (Mobley et al., 2016).

$$L_t = L_{atm} + L_{TOA\_surf} + L_{TOA\_w})$$

The principal difference between the atmospheric corrections of an image from a satellite sensor or one from an aircraft sensor is the incident solar light scattered and absorbed by some atmosphere molecules, such as ozone. The flight height of the aircraft is between 1 and 2 km, and it is not affected by ozone, because it is found in a layer whose height ranges between 20 and 30 km ([Dominguez et al., 2009]).

Techniques based on remote sensing imaginery have been applied to map water depth in cases of clear and shallow water (Lyzenga, 1978), (Eugenio et al., 2015). The use of active and passive optical sensors has allowed developing methodologies to map water depth in highly turbid waters (Sanchez-Carneroa, 2014).

Traditionally, bathymetric methods have been based on sampling data by defining the planimetric position of the point and applying a technique to measure the thickness of the water layer. Nevertheless, sometimes the working areas are not accessible
or they are very hard to directly sample, since they are death traps.

Nowadays, bathymetries are made with Global Positioning System (GPS) in Real Time Kinematic (RTK) mode, and Echo-sounder data, reaching higher accuracies than before. However, bathymetric works with traditional sounding techniques are slow, expensive and even dangerous. The feasibility of obtaining bathymetric charts from satellite or aircraft imagery has been demonstrated by several researchers and, in recent years, it is becoming increasingly interesting, because this kind of
information has the advantages of being available and having reasonable prices.

Accuracy in depth estimation is highly dependent on the atmospheric correction made to the imagery (Bayarri and Castillo, 2005). The reduction of effects such as glint and cross track illumination (CTI) in shallow water areas with homogeneous bottoms and water areas improve the results of the depth estimation. Several methodologies have been developed to remove the effect of atmosphere on the recorded sensor signal. Radiative Transfer Codes (RTC), such as raw data or raw data minus
band 34 (Edwards, A.J., 1999), 6S (Vermote et al., 1997) and 6S improved (Catalonian Cartographic Institute (ICC)) for water depth estimation have been widely used for this purpose (Castillo et al., 2011).

Water reflectance is related with water quality data (Secchi depth (SD), suspended solids concentration (TSS) and chlorophyll-a concentration ([Cla]). Considering the work by Dominguez, J.A. et al. (2009), in which ASD FieldSpec FR spectroradiometer results were calibrated using a 25% grey card reference panel (Goodin et al. 1993, Mayo et al. 1995, Hand and Rundquist

1998), it concluded that reflectance values cannot be greater than 1.5% because otherwise, the bottom would not be seen. Hence the importance of a good atmospheric correction.

The aim of this work is to assess four atmospheric correction methods to accurately determine the depth in the Bay of Santander (Cantabria, Spain), highlighting the importance of the atmospheric correction for airborne hyperspectral remote sensing in shallow waters.

## 2 Material and Methods

### 2.1 Study Area

The Bay of Santander is located in the North of the Spain, about 200 km away from its border with France. It is a depression with a very rich ecosystem from both biological and socio-economical points of view. More than 250,000 inhabitants are concentrated in this area, which means more than 50% of the population of Cantabria, the autonomous community to which it belongs.

Human spills aimed to convert sea into land started circa 1850, and they have gradually changed the tide prism. As a consequence of the coastal dynamics and the mouth of the Miera River, a singular structure called "el Puntal de Somo" has appeared (Figure 1). It is an approximately 2.5 km long and 250 m wide sand tongue. This made the Bay start filling up and, consequently, it produced navigation safety-related problems. Hence, the navigational channel must be periodically measured and dredged. The water of the Bay of Santander is clear and it is continuously monitored.

### 2.2 Remote Sensing Data

The imagery used in this work was taken by a CASI-2 sensor, owned by the Catalonian Cartographic Institute (ICC). CASI-2 is a pushbroom imaging spectrograph with a two-dimensional CCD array of 512 spatial pixels and 288 spectral pixels, which scans the scene in the VNIR (405-950nm). It allows the user to set up the number and width of the bands in which the sensor will record data.

The sensor was installed on board of the plane "Cessna Citation I", which belongs to the ICC. The flight and field data campaign were acquired by the ICC and the hour and date were set due to both high sun elevation (high penetration capacity) and low astronomical tide.

The flight parameters were designed according to the necessities (spatial resolution 4 x 4 $m^2$; integration time of 32 ms; speed of 121,5 knt; height of 5692 ft and a flight direction of approximately 135° in order to reduce the glint effect by flying with the sun on the front or on the back).

10 tracks were developed to capture the whole bay. The ICC, with 36 channels of 18 nm and a spatial resolution of 2 meters, sets the spectral configuration of the images. Besides, the secchi disk was applied as an indicator of transparency. 23 samples were acquired during the day of the flight.

The Center for Environmental Research of the Government of Cantabria (CIMA) produced a report on pollutants and meteorological parameters (PM10, $SO_2$, CO, NO, $NO_2$ and $O_3$) with the data from two stations of the Air Quality Control and Monitoring Network in Cantabria which are sited in the center of Santander.

The geometric correction of the imagery was made by using the sensor orientation data, the inertial system SISA, and a Digital
Terrain Model with a pixel size of 25 m (DTM25), which was obtained from the National Topographical Map 1:25.000, with a pixel interpolation with the nearest neighbor method.

Four variations of the imagery were considered according to the atmospheric level correction:

- SC: Raw data were simply corrected with the parameters of the annual calibration certificate, and resampled to 16 bits. Such corrections are used when working with airborne sensors, which do not have an incident light sensor (ILS) that
measures the color spectrum of incident light from the sun.

- NC – B34: Band 34 was subtracted from bands 1 to 34. Since works were held over a subtidal area, the Near Infrared (NIR) energy is mostly absorbed by water. Their values should be very close to zero. High values are mainly due to atmosphere, scattering and glint effects. The image brightness can be reduced by subtracting a NIR band to the visible bands. This has been done by subtracting band 34 (908-924 nm) to bands 1 to 24 (408-770 nm).

- C1: 6S (Second Simulation of the Satellite Signal in the Solar Spectrum) correction with default parameters and glint subtraction after 6S correction. The interface effects were corrected according to the empirical baseline adjustment developed by Silió-Calzada in 2002, which includes sun-glint and sky-glint removal.

- C2: Improved 6S correction developed by the ICC and the Government of Spain, considering in situ radiometric samples provided by spectroradiometer ASD-FR and pollutants and meteorological parameters of CIMA.

**2.3 Calibration and Validation Data using GPS in RTK mode and echo sounder.**

Data used to calibrate and validate the depths estimated by remote sensing were obtained by using a sonar installed on a boat (Figure 2-(c)). The main device is an Atlas Elektronic Deso 15, requiring different attachments, such as a foot tube fixed to

the GPS antenna, batteries and an alternating current generator to feed the instruments, and a laptop to store the measured data (Figure 2-(a), (b)).

The process followed implied measuring the depth with the echo sounder and, simultaneously, the absolute position with GPS. In this way, both data can be integrated to calculate a depth value with respect to the origin, which in this case is the mean sea level at Alicante.

Five hours were necessary to complete the depths sampling. The maximum browsing speed was 6 km/h, so as to get an accurate GPS-echo sounder data synchronization and thus transform the heights to the depths with respect to the reference origin. The measuring process was limited by a minimum depth of 1m, as the boat needed at least that depth to navigate. Due to this fact, it sailed through the navigation channels mainly.

**2.4 Bathymetric Algorithms**

Nowadays, the majority of the bathymetric works are developed by means of dual frequency GPS and echo sounder, obtaining very good accuracies. However, bathymetric measures in shallow waters using traditional surveying techniques are slow, expensive, and even sometimes dangerous. Hence the interest in obtaining depth estimations by means of satellite or airborne imagery. The bathymetry obtained by this methodology has advantages and disadvantages. The main advantages are that this information is available for most areas, and the prices are rather reasonable. Unfortunately, the precision obtained is not as good as that provided by GPS and echo sounder.

Classical algorithms for depth estimation when using airborne images involve the inversion of upwelling radiance, or some parameter derived from it, to recover depth. The problem is that at-sensor radiance measured over water is a function of the atmosphere, the clarity of the water column (which mainly depends on the chlorophyll, turbidity and organic matter), the bottom type and the water depth. For example, it can be particularly difficult to decouple the effects on upwelling radiance caused by changing bottom types from those caused by changing depths. Therefore, many depth algorithms require the knowledge of a few accurately measured depths or bottom characteristics for calibration. When these calibrating measurements are available, depths computed from spectral data can provide quite accurate information about the depth.

The classical bathymetric methods used to derive depth information emanate from simple regressions between the depth and a value calculated from a visible range wavelength band, Principal Component Analysis (PCA) or Richards and Karhunen-Loeve Transform. All these algorithms were implemented by using Interactive Data Language (IDL). Polcyn and Lyzenga developed a simple water reflectance model, which accounts for the major part of the signal received by a multispectral scanner over clear shallow water, but neglects the effects due to scattering in the water, and internal reflection at the water surface. A few years later (Lyzenga,1978) amended this model, including the effects of dispersion in water and the internal reflection of

the water surface. The model establishes that the dispersion has the same dependence on the depth that the radiance reflected from the bottom.

However, the model presented by Benny and Dawson is very simple because it establishes that the clear water in shallow areas allows the reflection of the light and it reaches the sensor. However, the amount of light that returns depends on the attenuation coefficient for that wavelength and the reflection coefficient of the bottom. Philpot's method is a radiative transfer model whose parameters depend on the wavelength, except for depth. The fundamental assumption of this model is that the optical properties of water are vertically homogeneous. This is not a hypothesis that fits the reality, but a starting point is required.

Finally, Hamilton method is a variation of Clark method, which allows the use of multiband analysis. It assumes that the background reflectance is constant. So, if this prediction model is applied together with profiles of irradiance attenuation, the effects of reflection of light in shallow bottoms can be reduced.

The following algorithms have been applied: simple regression; Polcyn & Lyzenga (two methods have been considered, using simple bands in the first case, and applying a range of the spectrum in the second, as the author suggests); Lyzenga has been also applied with the modifications made by Yarbrough and Easson; Benny & Dawson D.; E Philpot; F. Hamilton; principal component analysis (algorithms of Richards and of Karhunen-Loève) and a linear or logarithmic relation with the depth.

These bathymetric algorithms were separately applied in two different tracks: 12 and 13. In each case, linear and logarithmic regression adjustments were applied between the top of atmosphere reflectance values (TAR) for each correction model and each classical bathymetric algorithm in the locations of the calibration points (n=100), and the values provided by GPS-echo sounder in RTK mode. In the same way, another 100-point sample was used for the validation of the bathymetric results.

All the bathymetric algorithms have been calibrated and validated by using samples taken with the echo sounder and dual frequency GPS (200 points for each phase). Figure 1 shows the points that have been used for calibration and validation of algorithms.

## 3 Results

### 3.1 Radiometric assessment of hyperspectral images

Data from airborne sensors have varying degrees of brightness, which depend on the angle, the viewing angle and altitude sensor, the azimuth between the sun and the plane, and the type of surfaceobserved. If this effect is not corrected, or at least reduced, these variations can hinder the use of these images with standard algorithms, and the interpolating methods may mask interesting features of low spectral amplitude.

Different points of view have been considered to assess the imagery:

- ▪ Calculation of the digital number differences, according to their position across the flight path (CTI). They can be due to atmospheric effects, glint, the bidirectional reflectance distribution function of the bottom, etc. Two zones have been studied: the first one corresponds to deep water and the second to shallow water. To simplify the results, only CASI bands 1, 4, 8, 12, 17 and 24 are shown in Figure 3. Bands 1 and 24 are extremes in the SC-B34 correction. 17 and 4 CASI bands correspond to red and blue wavelengths respectively, and bands 8 and 12 are the extremes of the bands suggested by Hamilton (1993). The line represents the best-fit 1st-degree polynomial to data. Ideally, the slope should be close to zero, which means that the digital numbers (DN) are similar across the track of the flight.

- ▪ Assessment of radiometry from an absolute point of view. Samples of different spectral signatures have been taken to generate statistics of each correction level and hence, to assess them. TAR in blue and green ranges corresponds to correction SC (Figure 4). On the other hand, C1 and C2 corrections presents higher TAR in the NIR area (bands 17 and 24) as Figure 6 and 7 represents. Correction SC-B34 has reduced glint effect by almost 90%, depending on the considered band (Figure 5). This correction offers worse results when working with the trails left by vessels. Therefore, depending on the correction method, they can be considered a significant source of error.

Correction C1 offers a very low dynamic range, as the TAR trends to group between 1200 and 1800 values. In the same way, the standard deviations of these images present very low values. On the other hand, correction C2 offers a better dynamic range, but this improvement has produced a considerable increase of the standard deviation in most of the cases, bigger than the original imagery. This augmentation in standard deviation is not advisable for bathymetric aims.

The drawn lines represent functions of first setting for the digital values observed along the path. Ideally, they should have a slope equal to zero, which means that the areas marked as homogeneous present similar DN along the whole flight trajectory.

The best results in shallow water have been achieved by the image with C2 correction. However, images offer the best results with a level of correction C1 in deep water.

**3.2 Depth estimation using the atmospheric correction methods**

The process has been applied to tracks 12 and 13, for all levels of correction and classical and robust methods. The track 12 is in the in the Bay, predominating areas of shallow water (with a response that can be clay, loam or sandy), which is murky and calm. Flashes produced by waves do not add a very important quantity of signal, except for trails of the boats that sail in the area.

On the other hand, track 13 corresponds to the area between the Bay and the open sea, with greater depths and rougher water, which causes a significant increase in the signal recorded by the sensor because of the flashes.

## 4 Discussion

The following conclusions may be drawn about the suitability of the sensors applied to estimate the bathymetries may be drawn from the experimental work that has been developed. The sensor covers the full spectral range used in bathymetric algorithms (450 to 650 nm), its band width is 15 nm, and the radiometric resolution is 16 bits. It means that the sensor is very suitable for this purpose, as it allows processing multi-band algorithms and calculating mean bathymetries in ranges of wavebands (i.e. 450 to 500 nm). However, image brightness, glint effect and atmospheric scattering are outstanding due to the pixel size and the radiometric resolution. Due to this, the radiometric and geometric corrections have to be rigorous and precise.

Four levels of correction have been applied to the images (SC, SC-B34, C1 and C2), and the study area was characterized by using two tracks with different morphological characteristics.

Both the correction methods and the different depth estimation procedures have been validated. Error values results between 3 methods (SC, SC-B34 and C1) have been calculated because correction C2 increases the standard deviation more than the original image (Tables 1 and 2). In the same way, an analysis of the errors for the best methods of each type and track has been performed, observing that the distributions of errors are practically normal and centered at 0 (Figures 10 and 11).

Once applied the different corrections, it can be concluded that images are not internally homogeneous with a level of correction C2. According to the band considered, the SC-B34 reduces the flash caused by the waves by almost 90%. This provides worse results when trying to eliminate the waves left by ships.

On the other hand, the correction C1 offers a very low dynamic range since, as it is shown in Fig. 4, 5, 6 and 7, the DN tend to group between 1000 and 3000 with very low values of standard deviation. However, the level of correction C2 offers a better dynamic range, but this improvement has led to a considerable increase of the standard deviation in most of the cases studied, larger than the original image. Increased deviation is not advisable for bathymetric purposes. That fact, combined with the lack of internal homogeneity, have led to discard the use of the image with a level of correction C2.

The results were 17 bathymetries with the different methods. Figure 8 shows the results of the bathymetry obtained by applying Benny and Dawson method, and Figure 9 shows a bathymetry with ROBUST PCA Karhunen-Loève method.

The work carried out by other authors does not establish the importance of the atmospheric correction depending on the type of water and the conditions (waves, stelae of ships, contaminants, etc.). These aspects have been considered in this work. In this sense, this work highlights the importance of C1 and SC-B34 correction in shallow waters.

## 5 Conclusions

On the light of the results, it can be concluded that:

- In heterogeneous shallow water, the model of atmospheric correction C1 offers good results, since the 67% of the methods offer the best results when this correction is applied.

- In deep water with an additive of noise produced by flashes of waves, the results provided by the atmospheric model C1 were not as good as those mentioned above. The best bathymetry has been obtained with the level of correction SC-B34 and the method of Hamilton (PCA minus band 2), with an average error of 1.69 m. The correction C1 is appropriate for areas of deep water, as the reduction of flashes has been better with the correction SC-B34.

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

LIST OF FIGURES

# LIST OF TABLES

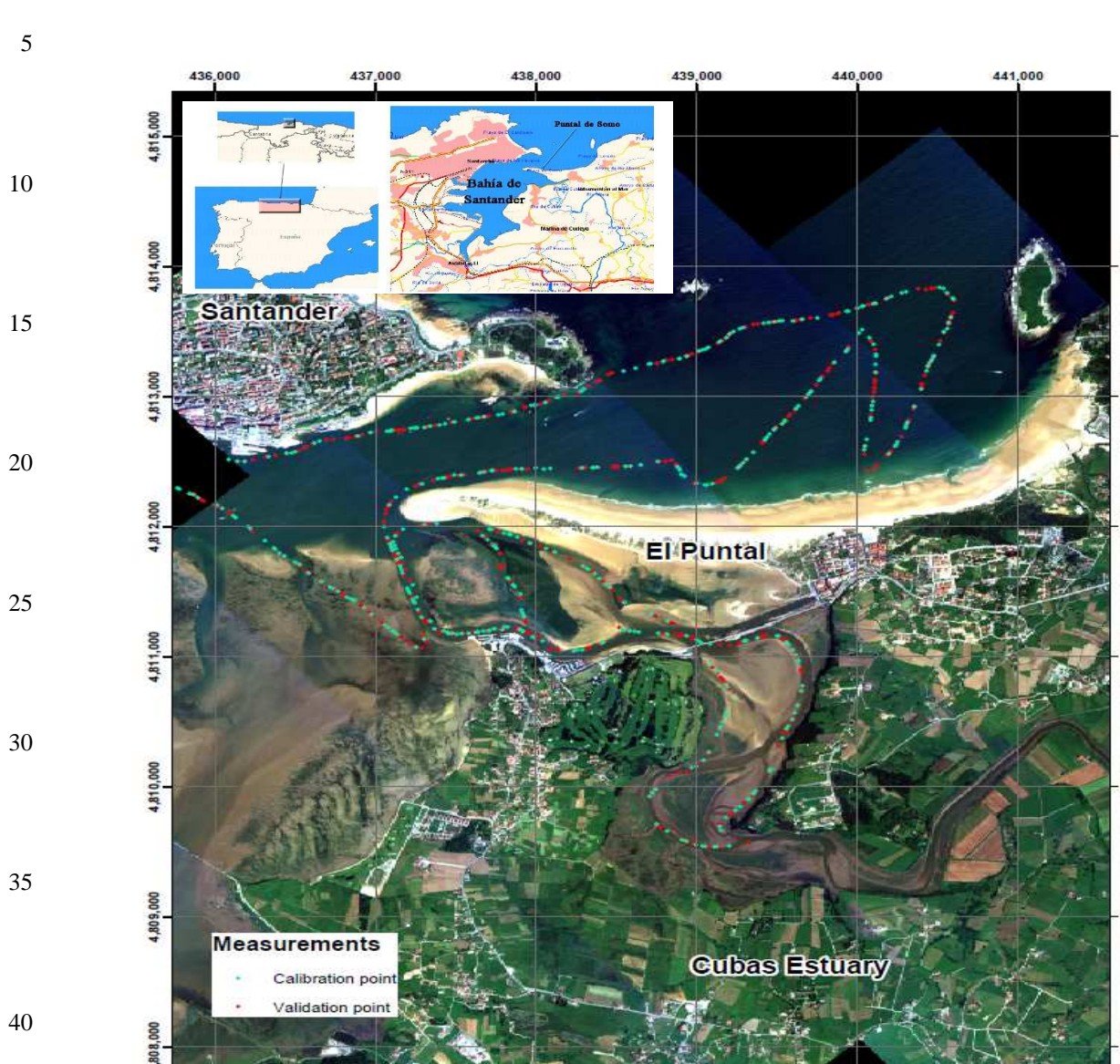

**Figure 1. Location of study sites in Santander Bay (North of Spain). Depth calibration and validation point with GPS in RTK mode and echo sounder.**

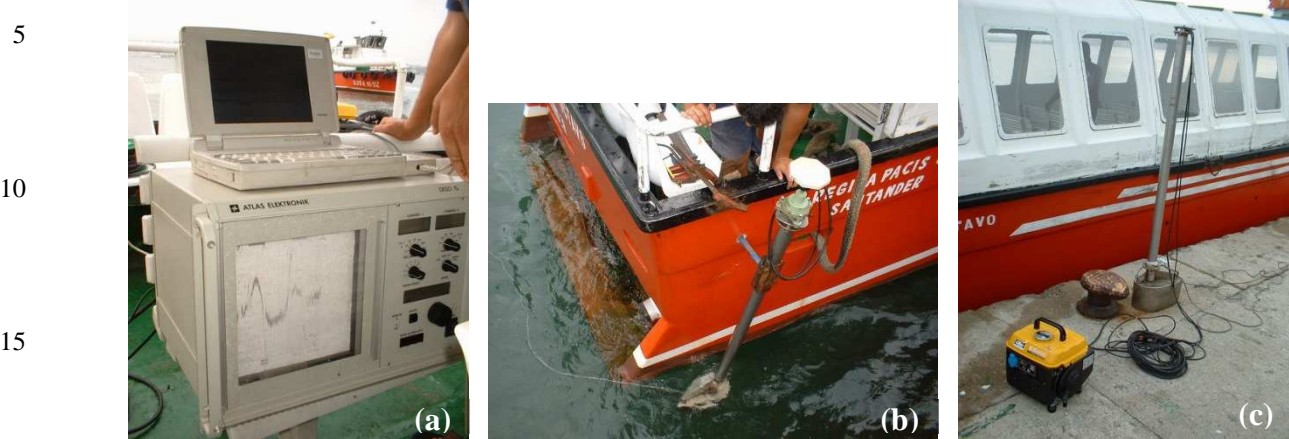

**Figure 2. (a) Echo-sounder and GPS in RTK mode. (b) Accessories. (c) Probe installed on boat.**

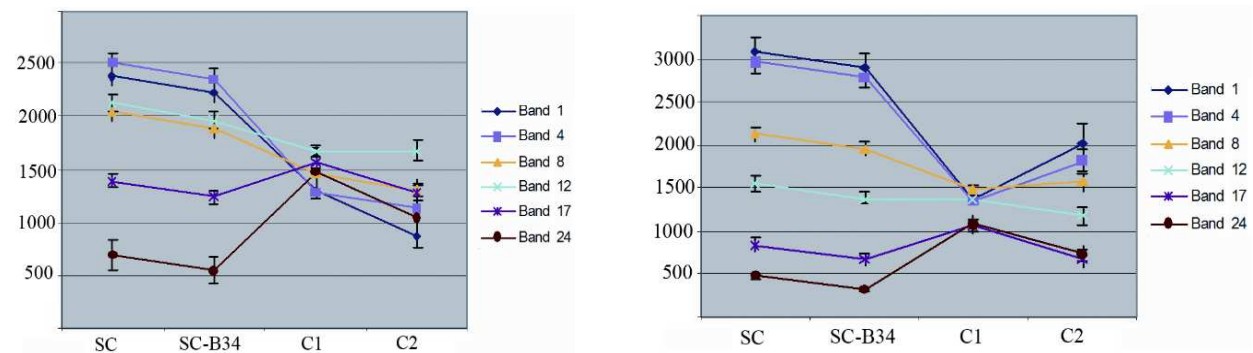

**Figure 3. (a) DN variation in deep water. (b) DN variation in the crest of the wave.**

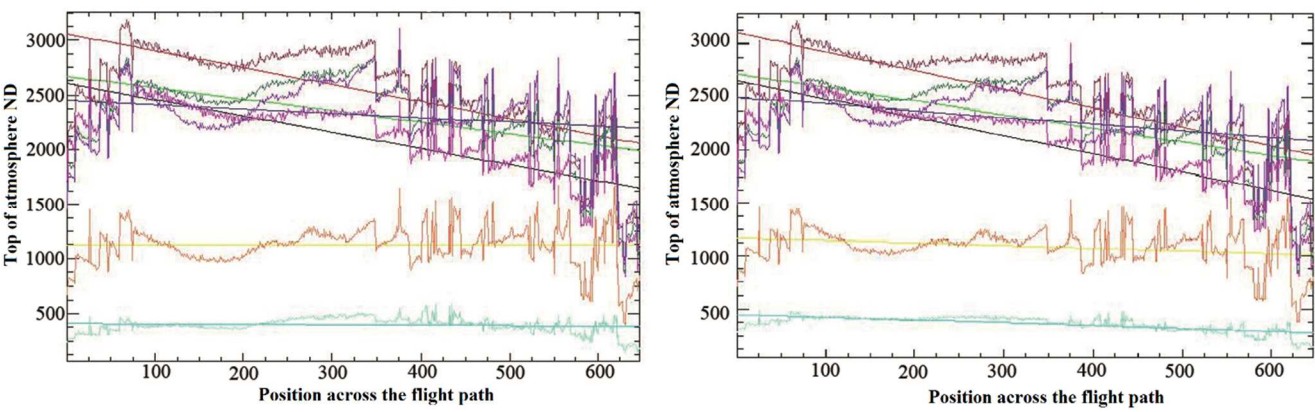

**Figure 4. (a) CTI in shallow water with SC. (b) CTI in deep water with SC.**

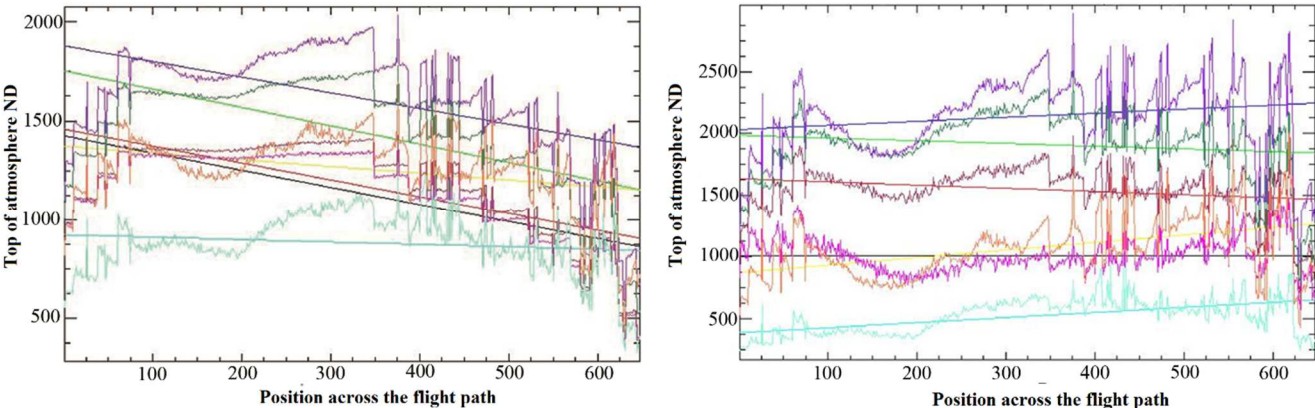

**Figure 5. (a) CTI in shallow water with SC-B34. (b) CTI in deep water with SC-B34.**

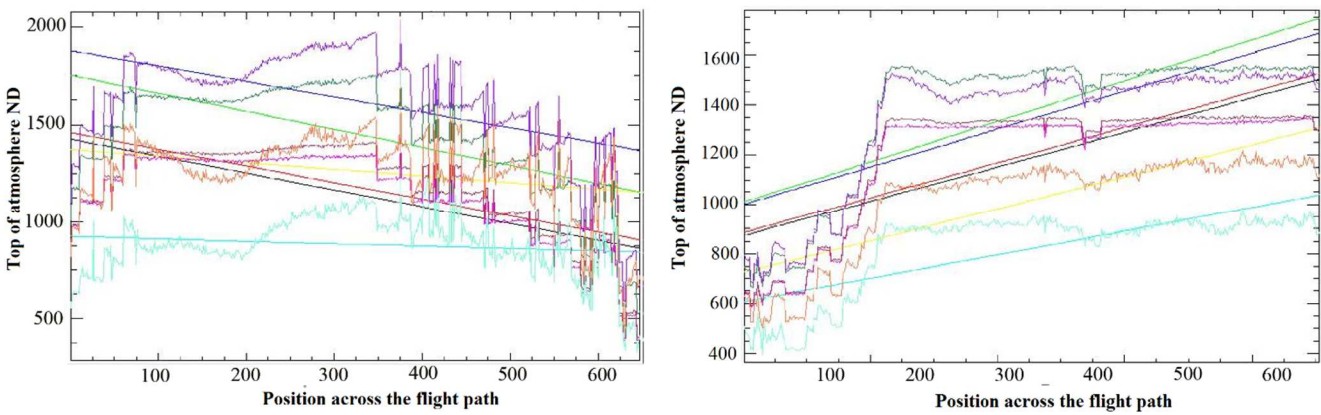

**Figure 6. (a) CTI in shallow water with C1. (b) CTI in deep water with C1.**

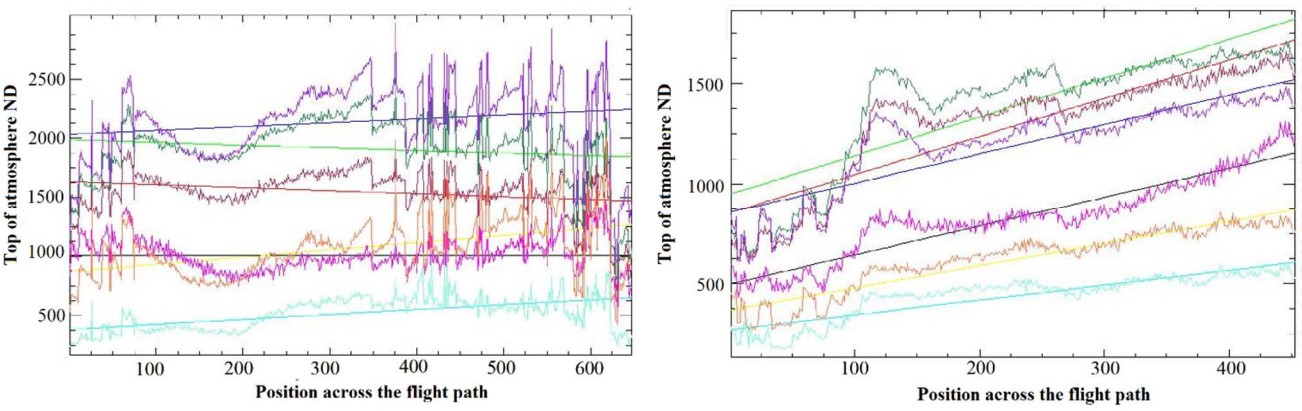

**Figure 7. (a) CTI in shallow water with C2. (b) CTI in deep water with C2.**

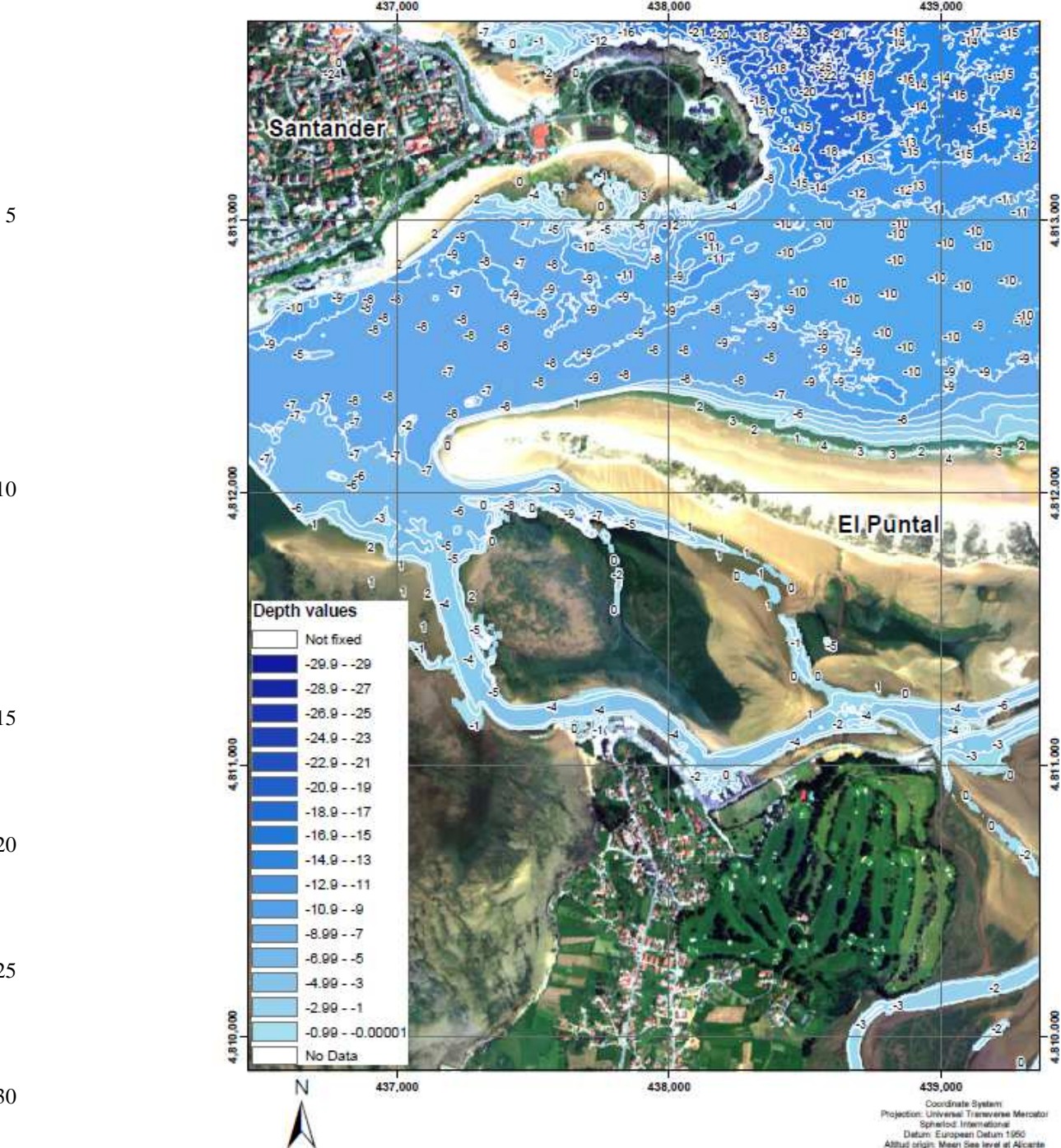

**Figure 8. Bathymetry with Benny & Dawson method**

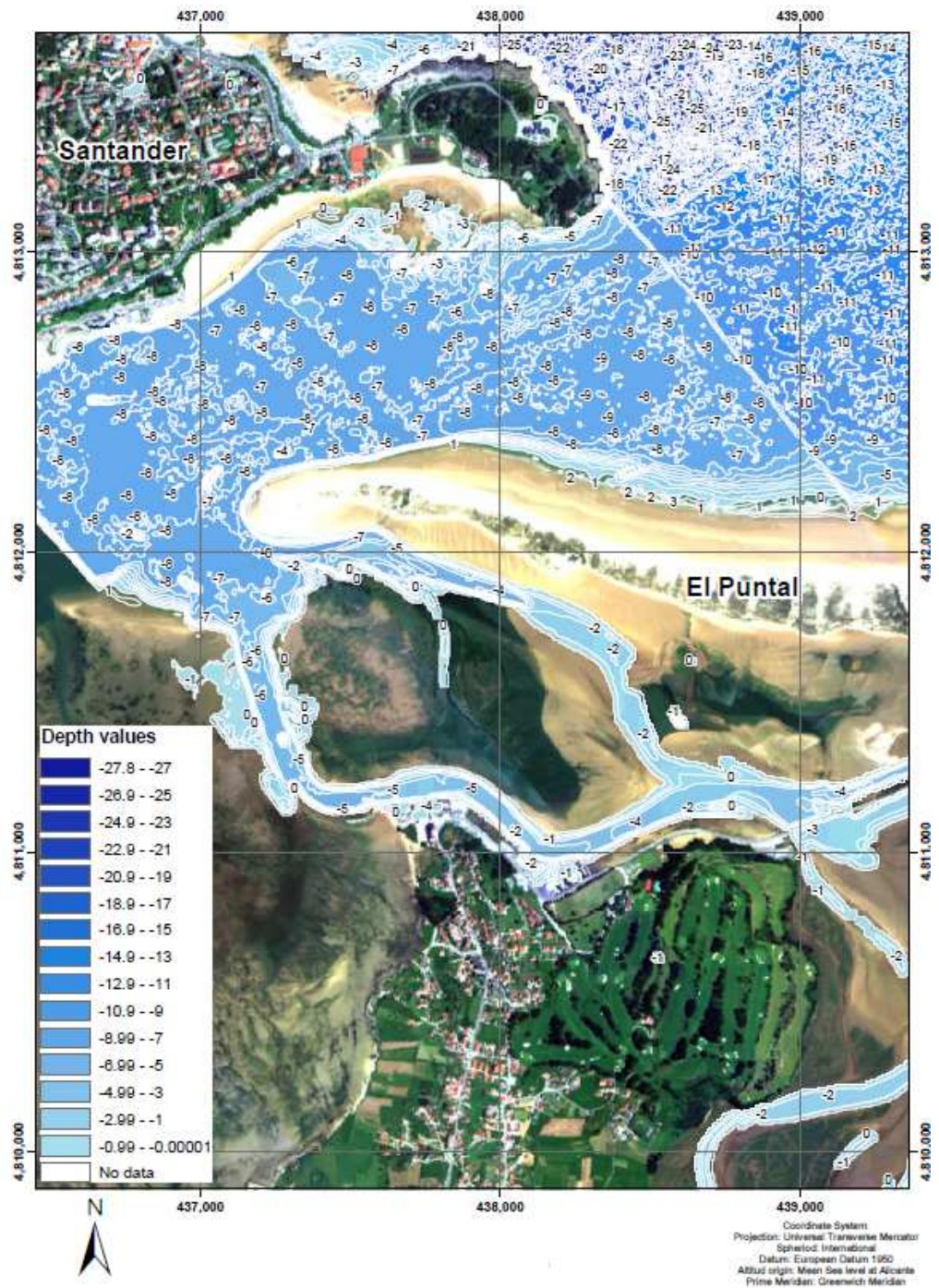

**Figure 9. Bathymetry with Robust PCA Karhunen-Loève method.**

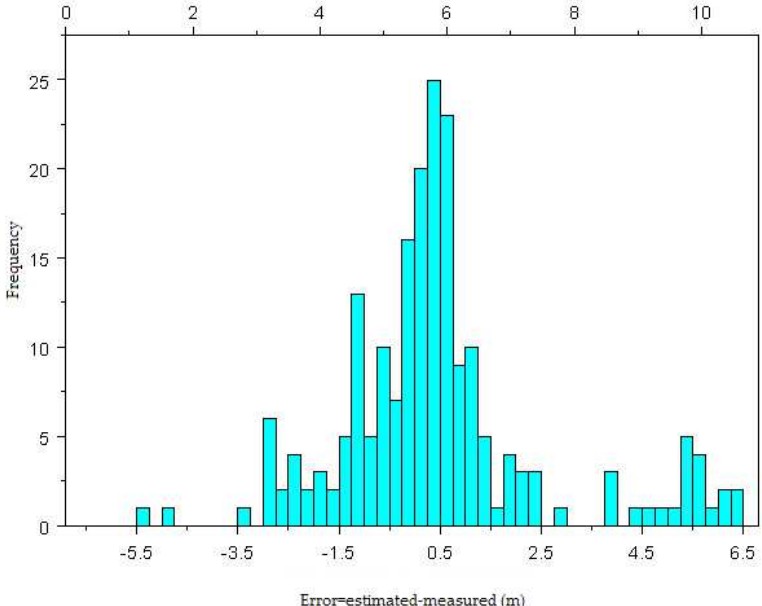

**Figure 10. Distribution of errors in Lyzenga method and correction level SC-B34 in track 12.**

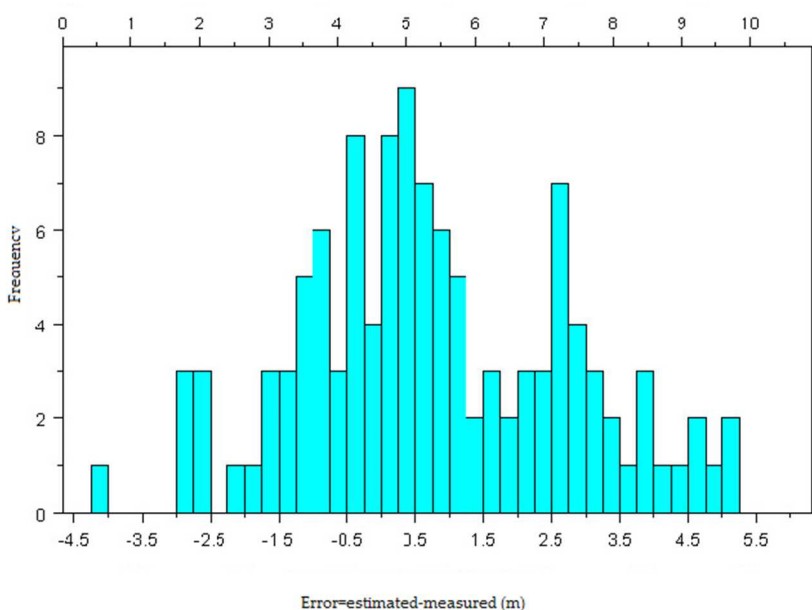

5    **Figure 11. Distribution of the errors in Hamilton method and correction level SC-B34 in track 13.**

**Table 1. Validation of the results for track 12 (shallow waters).**

| Track 12 | SC | | SC-B34 | | C1 | |
|---|---|---|---|---|---|---|
| Units (meters) | **Mean** | **Std Dev** | **Mean** | **Std Dev** | **Mean** | **Std Dev** |
| **Regresión.Simple** | 1.909 | 2.403 | 2.388 | 2.584 | 1.926 | 2.491 |
| **Polcyn & Lyzenga** | 1.900 | 2.657 | 1.995 | 2.575 | 1.813 | 2.609 |
| **Lyzenga** | 2.689 | 3.527 | 2.765 | 3.616 | 2.176 | 3.055 |
| **Hamilton** | 1.498 | 2.161 | 1.735 | 2.254 | 1.502 | 2.106 |
| **Philpot** | 1.590 | 2.325 | NA | NA | 1.633 | 2.325 |
| **PCA-KL** | 2.028 | 2.931 | 3.233 | 4.206 | 2.112 | 3.051 |

**Table 2. Validation of the results for track 13 (deep waters).**

| Track 13 | SC | | SC-B34 | | C1 | |
|---|---|---|---|---|---|---|
| Units (meters) | **Mean** | **Std Dev** | **Mean** | **Std Dev** | **Mean** | **Std Dev** |
| **Regresión.Simple** | 2.388 | 2.584 | 4.405 | 3.248 | 3.176 | 3.151 |
| **Polcyn & Lyzenga** | 1.995 | 2.575 | 2.417 | 2.707 | 2.349 | 2.896 |
| **Lyzenga** | 2.765 | 3.616 | 2.740 | 3.590 | 4.279 | 5.046 |
| **Hamilton** | 1.735 | 2.254 | 1.690 | 2.206 | 1.813 | 2.353 |
| **Philpot** | NA | NA | 1.999 | 2.513 | 1.942 | 2.536 |
| **PCA-KL** | 3.233 | 4.206 | 2.428 | 3.203 | 3.476 | 4.355 |