# Peer review of "The importance of Atmospheric Correction for Airborne Hyperspectral Remote Sensing of Shallow Waters. Application to Depth Estimation."

_Atmospheric Measurement Techniques, 2017_

## Referee Comment (RC1) · Anonymous Referee #3 · 31 May 2017

<General Comments>

The topics is interesting and from an engineering point of view, the technique is important. I have the following general comments

(1) The authors mentioned "to assess the best atmospheric correction method". 4 methods are described in page 4. However, error values between 4 methods are not well described.

(2) Geometry of the sun position, viewing direction, and observation point is critical for radiative transfer but it is not described well.

[Figure]

(3) Scope of AMT is measurement of gases, aerosols, and clouds of the Earth's atmosphere. But the paper describe atmospheric correction

(4) Estimated water depth and validated data should be compared clearly and presented in a figure or a table.

(5) Figures and their captions are not consistent. I recommend to submit another journal or resubmit the manuscript by describing radiative transfer of the water and atmosphere and estimating error analysis in more detail.

<Specific Comments>

(1) Page 2, line 25, ozone ozone absorption has strong spectral dependency.ãĂĂThe incident solar light to the surface is also affected by ozone absorption.

(2) Page 4, line 8, incident light sensor (ILS) Description of the incident light sensor is needed.

(3) Page 7, line 14, NNDD Definition of NNDD should be described.

(4) Page 12, Figure 3, Y axis "top of atmosphere reflectance": Usually reflectance is between 0 and 100%. Values look strange. The caption is "NNDD" but figure shows reflectance.

(5) Page 13, Figure 4 The figure shows NNDD but the caption is "CTI".

(6) Pages 13 and 14, Figures 4, 5, 6, and 7 What is the unit of CTI? Definition or description of X axis "range" is needed. Explanation of linear lines is needed.

<Technical Corrections>

(1) Page 2, line 10, Ku=Kd > "Ku= -Kd" or Line 9 Eq Kd-KFu > "Kd+Ku"

(2) Page 12, figure 2 There is no (a) (b) and (c) in the figure. The second photos looks like (c).

---

## Author Comment (AC1) · 7 Jun 2017

Dear Anonymous Referee #3,

I am writing to thank you for the comments and suggestions you provided me about the manuscript titled "The importance of Atmospheric Correction for Airborne Hyperspectral Remote Sensing of Shallow Waters. Application to Depth Estimation". I would like to thank the time you devoted to the revision of the document.

In relation with the aspect that you pointed me, "to assess the best atmospheric correction method", I have expanded this point, error values results between 3 methods (SC, SC-B34 and C1) because correction C2 offer a better dynamic range but increase the standard deviation larger than the original image. (Page 7, lines 21 to 25, Tables 1 and 2, Figures 10 and 11).

In relation with the geometry of the sun position, viewing direction and observation point, I have expanded the description as you can see in Page 4, lines 1 to 10.

This manuscript is a research article with significant advances in remote sensing using in situ and laboratory measurement techniques with detailed error analysis. Information retrieval for gases, aerosols, and clouds are an important part of this article for atmospheric correction of the imagery (Page 4, line 11 to 31) and for depth estimation in shallow waters because the accuracy in bathymetric information is highly dependent on the atmospheric correction made to the imagery. The reduction of effects such as glint and cross track illumination in homogeneous shallow-water areas improves the results of the depth estimations.

I have made all corrections you posed me in the specific comments and I have corrected an error in the document regarding the acronym NNDD that refers to the digital number (DN). The main document has been modified in this sense.

I remain at your disposal for any further comments or suggestions about it.

Yours faithfully,

Elena Castillo-Lopez

Please also note the supplement to this comment:
http://www.atmos-meas-tech-discuss.net/amt-2017-37/amt-2017-37-AC1-supplement.pdf

―――――――――――――――――

---

## Referee Comment (RC2) · Anonymous Referee #4 · 21 Jul 2017

Review of the manuscript amt-2017-37 "The importance of Atmospheric Correction for Airborne Hyperspectral Remote Sensing of Shallow Waters. Application to Depth Estimation" by E.Castillo-López, J. A. Dominguez, R. Pereda, J. M. de Luis, R. Pérez, F.Piña.

Author presented their research on applying airborne remote sensing imagery for water depth estimation. They show some interesting results, however, the quality of manuscript requires quite substantial improvement before it can be adequately peer-reviewed. The manuscript is in very rough shape and is not suitable for detailed scientific and technical review in its present form. Also, it does not really present, in my opinion, any significant novel scientific results related to atmospheric measurement techniques, i.e. the major subject of the AMT journal. My recommendation is to return manuscript back to authors for further improvements and possible re-submission to another journal focused more on the bathymetry measurement techniques.

Major comments:

1) English style and grammar, as well as clarity of presentation must be improved. Manuscript generally does not follow the style required for AMT publication.

2) The math expressions contain misprints (errors). Not all notations are properly introduced and explained, they are not always properly formatted and consistent throughout the text. I would suggest to use the Math Editor for equations rather than regular text.

3) References are placed in somewhat arbitrary way. Some references are cited (such for example Pereda et al. 2016) but not included in the text. Some references are misleading. For example, Vahtmäe et al. (2006) and Castillo et al., (2011) are not authors of 6S radiative transfer code. Authors of 6S code (Vermote et al ) are not cited in the text. Mishchenko et al. (2004) work deals with analysis of measurement requirements for monitoring of aerosol forcing of climate from space, not with the issues "raw data and raw data minus band 34" as described by the authors.

4) Figures are generally of low quality. Not all figures are properly discussed in the manuscript.

5) Discussion of reflectances is somewhat confusing. Numbers looks strange -1200-1800 etc. NNDD notation is not explained.

6) Sections "Discussion" and "Conclusions' do not look very strong and convincing.

---

## Author Comment (AC2) · 11 Aug 2017

Dear Anonymous Referee #4,

I am writing to thank you for the comments and suggestions you provided about the manuscript entitled "The importance of Atmospheric Correction for Airborne Hyperspectral Remote Sensing of Shallow Waters. Application to Depth Estimation". I would like to thank the time you devoted to the revision of the document.

Firstly, English grammar and style have been reworked. In relation with the aspect that

you pointed, "Also, it does not really present, in my opinion, any significant novel scientific results related to atmospheric measurement techniques, i.e. the mayor subject of the AMT journal", I have expanded this point because I think the objective of the work was not been well explained in the previous version. It is intended to be able to apply this type of information (hyperspectral imagery) to the study of bathymetries where the reflectance values cannot be greater than 1.5% because otherwise, the background would not be seen. Here lies the importance of a good atmospheric correction. In some cases, the atmospheric correction itself is more important than the bathymetry measurement technique applied.

Water reflectance is related with water quality data (Secchi depth (SD), suspended solids concentration (TSS) and chlorophyll-a concentration ([Cla]), as Dominguez, J.A. et al. showed in the article "Monitoring transparency in inland water bodies using multispectral images) where ASD FieldSpec FR spectroradiometer results were calibrated by the use of a 25% grey card reference panel (Goodin et al. 1993, Mayo et al. 1995, Hand and Rundquist 1998).

Figure 1. Water reflectance spectra with an unknown maximum at 386 nm, [Cla]<1 mg m-3. (Dominguez, J.A. et al, 2009).

Figure 2. Water reflectance spectra with an unknown maximum at 386 nm, [Cla]>1 mg m-3. (Dominguez, J.A. et al, 2009).

Figure 3. Water reflectance spectra showing a minimum at 680 nm corresponding to the maximum absorption of chlorophyll-a and a relative maximum at 705 nm corresponding to the fluorescence of chlorophyll-a. (Dominguez, J.A. et al, 2009).

This manuscript is a research article with significant advances in remote sensing, using in situ and laboratory measurement techniques with detailed error analysis. Information retrieval for gases, aerosols, and clouds are an important part of this article. Considering the case of shallow waters, the accuracy in bathymetric information is highly dependent on the atmospheric correction made to the imagery. The reduction of ef-

fects such as glint and cross track illumination in homogeneous shallow-water areas improves the results of the depth estimations.

In relation with the points (1) and (2) of the Major comments I have improved the document in order to follow the style required for AMT publication. Moreover, I have used the math editor and I have introduced and explained the math expressions. In relation with point (3) I have made a revision of the references cited and I have changed the references about removal of scattering in the atmosphere and external reflection from the water surface. Page 3 lines 25 to 26.

To be able to create bathymetric maps, the effects that electromagnetic radiation (EMR) suffers must be taken into account. As the EMR reaches the earth's surface, molecules and particles of the land, water and atmosphere environments interact with solar energy in the 400- 950 spectral region that CASI receives through absorption, reflection, and scattering. This article compares different atmospheric correction methods, as it analyzes the importance of the air column beyond the method applied to estimate the depths. Data from airborne sensors are known to have varying degrees of angle dependent brightness variation, which change with the sensor view angle and altitude, the angle between the sun azimuth and the scan sensor plane, and the surface type of the land. If this effect is not corrected or at least reduced, these variations can hinder the use of data with standard image processing, and the interpretation methods may mask low-amplitude spectral features of interest.

The Near Infrared (NIR) energy is mostly absorbed by water. Their values should be very close to zero. High values are mainly due to atmosphere, scattering and glint effects. As Mischenko M. and Dubovik O. suggested to the author in "Advances in light scattering by particle systems" ( an International Workshop that took place in Laredo (Spain), from 5 to 7 of July, 2004), the image brightness has been reduced by subtracting a NIR band to the visible bands. The image brightness effect reduction has been done by subtracting band 34 (908-924 nm) to bands 1 to 24 (408-770 nm) which have been used in the bathymetric algorithm.

The raw data minus band 34 is useful if there is not access to atmospheric models, such as the 6S radiative transfer codes.

The work carried out by other authors does not establish the importance of the atmospheric correction depending on the type of water and the conditions (waves, stelae of ships, contaminants, etc.), in which we are working. In this sense, this work highlights the importance of C1 and SC-B34 correction in shallow waters.

About the point (4), I have tried to improve the quality of the figures and to make a better discussion in the manuscript about it.

I have corrected an error in the document regarding the acronym NNDD, that refers to the digital number (DN). The main document has been modified in this sense while making the corrections that Referee #3 suggested. Finally, in order to make the Dicussion and Conclusions more strong and convincing, I have expanded these sections. I have also expanded the description about the sun position, considering direction and observation point, as it can be found in Page 4, lines 25 to 27 and Page 5 lines 1 to 11). Regarding the aspect "to assess the best atmospheric correction method", error values results between 3 methods (SC, SC-B34 and C1) have been introduced, as correction C2 offers a better dynamic range, but it increases the standard deviation more than the original image.

I am available for any further comments or suggestions about this work.

I look forward to hearing from you soon.

Yours faithfully,

Elena Castillo

Please also note the supplement to this comment:
https://www.atmos-meas-tech-discuss.net/amt-2017-37/amt-2017-37-AC2-supplement.zip

[Figure]

[Figure]

Fig. 1.

[Figure]

(chart)

R_w (%) vs Wavelength (nm)

—[Cla]= 1.14 mg m$^{-3}$ [TSS]= 1.88 mg L$^{-1}$ SD= 4.27 m   —[Cla]= 1.16 mg m$^{-3}$ [TSS]= 2.10 mg L$^{-1}$ SD= 4.10 m

—[Cla]= 1.45 mg m$^{-3}$ [TSS]= 1.54 mg L$^{-1}$ SD= 4.18 m   —[Cla]= 1.93 mg m$^{-3}$ [TSS]= 1.72 mg L$^{-1}$ SD= 3.32 m

—[Cla]= 2.78 mg m$^{-3}$ [TSS]= 1.24 mg L$^{-1}$ SD= 3.43 m

**Fig. 2.**

[Figure]

Legend:
— [Cla]= 10.67 mg m$^{-3}$ [TSS]= 7.98 mg L$^{-1}$ SD= 1.19 m
— [Cla]= 11.89 mg m$^{-3}$ [TSS]= 6.62 mg L$^{-1}$ SD= 1.04 m
— [Cla]= 13.47 mg m$^{-3}$ [TSS]= 12.14 mg L$^{-1}$ SD= 1.11 m
— [Cla]= 14.60 mg m$^{-3}$ [TSS]= 7.50 mg L$^{-1}$ SD= 1.0 3 m
— [Cla]= 14.82 mg m$^{-3}$ [TSS]= 8.94 mg L$^{-1}$ SD= 1.08 m

**Fig. 3.**